# An Internet-Based Multi-Approach Intervention Targeting University Students Suffering from Psychological Problems: Design, Implementation, and Evaluation

**DOI:** 10.3390/ijerph19052711

**Published:** 2022-02-25

**Authors:** Marlene De Fabritiis, Federica Trisolini, Gloria Bertuletti, Ionut Daniel Fagadau, Davide Ginelli, Katiuscia Pia Lalopa, Lisa Peverelli, Alessia Pirola, Gaia Sala, Marta Maisto, Fabio Madeddu, Jorge Lopez-Castroman, Daniele Romano, Alessandro Gabbiadini, Emanuele Preti, Daniela Micucci, Raffaella Calati

**Affiliations:** 1Department of Psychology, University of Milano-Bicocca, 20126 Milan, Italy; m.defabritiis@campus.unimib.it (M.D.F.); f.trisolini@campus.unimib.it (F.T.); g.bertuletti1@campus.unimib.it (G.B.); k.lalopa@campus.unimib.it (K.P.L.); l.peverelli2@campus.unimib.it (L.P.); a.pirola14@campus.unimib.it (A.P.); g.sala50@campus.unimib.it (G.S.); m.maisto2@campus.unimib.it (M.M.); fabio.madeddu@unimib.it (F.M.); daniele.romano@unimib.it (D.R.); alessandro.gabbiadini@unimib.it (A.G.); emanuele.preti@unimib.it (E.P.); 2Department of Informatics, Systems and Communication (DISCo), University of Milano-Bicocca, 20126 Milan, Italy; i.fagadau@campus.unimib.it (I.D.F.); davide.ginelli@unimib.it (D.G.); daniela.micucci@unimib.it (D.M.); 3BICAPP—Bicocca Center for Applied Psychology, University of Milano-Bicocca, 20126 Milan, Italy; 4Department of Adult Psychiatry, Nîmes, University Hospital, 30900 Nîmes, France; jorge.lopezcastroman@chu-nimes.fr; 5IGF, CNRS-INSERM, Université Montpellier, 34094 Montpellier, France; 6CIBERSAM, 28029 Madrid, Spain

**Keywords:** internet-based intervention, cognitive-behavioral treatment, university students, randomized controlled trial

## Abstract

Despite the widespread prevalence of mental health problems, most psychological distress remains untreated. Internet-based psychological interventions can be an essential tool for increasing treatment availability and accessibility. The main objective of the MindBlooming project is to design and implement an innovative Internet-based multi-approach treatment for university students suffering from psychological or physical problems. The intervention will focus on symptoms of depression, anxiety, sleep problems, self-destructive thoughts, job- and study-related stress and burnout, and chronic pain. It will be based on different approaches, primarily psychoeducation, Cognitive-Behavioral Treatment (CBT), and third-wave CBT. At the end of the treatment, user satisfaction and usability will be assessed. In addition, two further aims will be evaluating the treatment efficacy through a randomized controlled trial and tuning a predictive model through Machine Learning techniques. The intervention consists of a 7-week treatment on two problematic areas according to each students’ personal needs, identified through an initial assessment. Besides the treatment assigned following the initial screening, participants will also be assigned to a different module to improve their relational skills. The treatment, which can be accessed through a mobile app, consists of psychoeducational videos followed by related exercises. We expect MindBlooming to be a remarkable tool for promoting the mental health of university students.

## 1. Introduction

Worldwide, data about the prevalence of mental health issues are alarming: in 2020, during the COVID-19 pandemic, more than 30% of the general population suffered from depression, anxiety, distress, or insomnia [1]. University students appeared to be highly vulnerable to the risk of mental health deterioration during this period [2]. Specifically, in Italy, between 2019 and 2021, 14% of university students had at least one mental disorder [1]. Young adulthood is a critical phase during which the first psychological symptoms may appear, triggered by the stress related to the university studies and to the new responsibilities of adulthood [2,3]. The prevalence of psychological issues among university students, including depression, anxiety [4], insomnia [5], suicidal thoughts and behaviors [6], stress or burnout [7], and chronic pain [8], appears to be increasing worldwide [5,9]. Mental well-being is connected to our emotionality [10], which is why social isolation and loneliness have a substantial impact on mental health outcomes [11,12] and emotional regulation difficulties represent a risk factor for mental disorders [13]. Furthermore, psychological distress in university students is often associated with a higher risk of failure and dropout from the course of studies [14].

Despite the psychological problems’ high prevalence and significant negative consequences (linked to disability, severe socio-economic issues, and even death), only a minority of university students can access proper treatment [15]. Consequently, promoting university students’ well-being should be a priority of the educational and social systems [16].

Internet-based interventions (IBIs) are a promising way to improve treatment accessibility and availability [17]. These entail the technological implementation of therapeutic treatments that occur in different ways: through web-based platforms, via mobile applications, virtual reality, video games, or feedback and reinforcement interventions through notifications and nudges [15]. IBIs overcome numerous limitations of more traditional forms of treatment related to costs, waiting lists, time and space limits, and the stigma associated with psychotherapy [18]. Several studies reported the effectiveness of self-help IBIs for different issues (i.e., depression, anxiety, sleep-related problems, and chronic pain) compared to control groups [15]. Comparable effectiveness to traditional face-to-face treatments has been observed [19,20]. To date, no risks or negative effects of IBIs have been highlighted [21]. IBIs appear to be a promising tool for university students because, among them, the use of the Internet and smartphones is widespread [22].

MindBlooming is a mobile app for a multi-approach treatment of psychological and physical problems in university students. It represents an attempt to respond to the growing psychological distress of the university population through an easily accessible and always available treatment. The name “MindBlooming” was chosen to evoke the idea of “mental growth”, framing one’s mind as a plant that needs to be cared for. Similar studies have already been performed worldwide [23,24,25,26] but most of them are based on a single intervention approach and face only one or two mental problems. MindBlooming offers a multi-approach and transdiagnostic treatment that allows users to work on different issues through the same app, reducing the commitment needed to use various applications for comorbid issues and therefore increasing treatment efficacy [27]. To the best of our knowledge, this will be the first attempt to deliver such an intervention (i.e., a multi-approach IBI that deals with different psychological and physical problems, and is tailored) to the university students’ population.

### Aims and Hypotheses

MindBlooming is a two-year research project with multiple primary goals and related hypotheses. Firstly, we aim to design and implement a new Internet-Based Multi-Approach Intervention (IB-MAI) for university students. The intervention covers a wide range of mental health issues that are relevant within the university student population [11,28] and that are treatable with IBIs [23,25,29]. Furthermore, the intervention is tailored to the individual’s needs. The mental health issues that MindBlooming addresses are depression, anxiety, sleep problems, self-disruptive thoughts, job- and study-related stress and burnout, and chronic pain.

Secondary aims will be: (1) the feasibility assessment of this IB-MAI (in terms of acceptability, satisfaction, usability, and uptake) through a pilot study on a restricted sample; (2) a randomized controlled trial (RCT) to compare the IB-MAI with web-education, as the control condition (CC); and (3) exploratory analysis to tune a predictive model on the collected data adopting Machine Learning techniques.

We hypothesize that: (1) the design and implementation of the IB-MAI treatment will be feasible; (2) the majority of the students will accept and be satisfied with the treatment; (3) the IB-MAI will be superior to the CC on both the primary and secondary outcome measures; and (4) the large amount of collected data will allow the identification of predictors of treatment response and different trajectories in treatment response.

## 2. Materials and Methods

The project has already been approved by the Internal Review Board (IRB) of the University of Milan-Bicocca at the beginning of the year 2021 (protocol number: 0018862/21; date: 16 February 2021). The study will be organized into two phases. The first phase, which has already been completed, focuses on the IB-MAI design and implementation. An in-depth analysis of current IBIs was conducted to design the IB-MAI. This allowed us to analyze the strengths and limitations of different approaches and use these observations to guide the development of the intervention. Our IB-MAI was designed as a mobile app for both Android and iOS to increase the number of potential users. MindBlooming provides a set of psychoeducational videos (with transcripts) about the psychological issues faced during treatment, as well as questionnaires and exercises to collect information before, during, and after the treatment itself. We followed the design principle of Material Design [30], which is a system of guidelines, components, and tools that support the best practices of user interface design and specify theories and practical applications of the interaction among human and machine for mobile development. The user interface (Figure 1) has been developed following the current usability standards, and it comprises features such as reminders, nudges, a calendar, and a diary.

The second phase will consist of 3 main steps. In the first step, a pilot study will be performed by recruiting 30 participants to evaluate the feasibility of this IB-MAI (acceptability and satisfaction, usability, and uptake). An RCT will be conducted in the second step. The pilot study and the RCT structure will be described in detail in the following paragraphs. Lastly, data analysis will be performed through Machine Learning techniques.

### 2.1. Participants and Recruitment

University students aged between 18 and 29 will be invited to participate; to be included, they must declare to have experienced at least mild- or moderate-intensity forms of mental health symptoms (no specific diagnostic screening will be performed in this phase), and they must not currently be in psychotherapy or start psychotherapy during the entire duration of the treatment. Participants are also required to have a phone based on Android or iOS operating systems and an Internet connection. Exclusion criteria include the inability to communicate in Italian.

Participants will be recruited through social networks (e.g., Facebook and Instagram), students’ WhatsApp groups (to include students not using the mentioned social networks, thus avoiding selection biases), and the Sona System recruitment platform made available by the University of Milan-Bicocca. Moreover, to further encourage participation, posters will be placed in local universities. In this phase they will receive preliminary information. Then, through a web link, they will be redirected to a specific landing page with more detailed information about the mobile app. On this web page, participants will be redirected to the proper marketplace (Apple Store and Google Play) to download the app on their smartphones. If possible, European Credit Transfer and Accumulation System (ECTS) credits will be awarded to participants. Once their eligibility has been verified and they have signed an online informed consent form, students interested in participating will be included in the study. We will collect the information concerning the diverse participants’ recruitment channels to ensure that there is no difference in baseline data among different recruitment channels. The same standards for data collection will be applied through different channels.

### 2.2. Pilot Study Design

For the pilot study, 30 participants will be recruited to evaluate the feasibility (in terms of acceptability and satisfaction, usability, and uptake) of the IB-CBT program. Inclusion and exclusion criteria, detailed in the following paragraphs, will be the same as for the RCT. Baseline and post-treatment assessments will be conducted using the same methods and scales used in the RCT and described in the following paragraphs. We assume most students will accept the treatment and be satisfied with it. However, some participants may run into technical or access issues, as stated by previous studies [28] (e.g., Wilks et al., 2018).

#### Feasibility Measures

At the end of the pilot, different feasibility aspects will be assessed using the following scales:❖Treatment satisfaction, measured by a modified version of the Client Satisfaction Questionnaire (CSQ-8) [31], an 8-item questionnaire rated on a 4-point Likert scale; and a Visual Analogue Scale (VAS) [32];❖System usability, via completion of a modified version of the System Usability Scale (SUS) [33], which is a 10-item questionnaire rated on a 5-point Likert scale that provides a global measure of subjective usability; and a VAS [32];❖Adherence rates for each lesson and the overall program.

### 2.3. RCT Design

This study will be conducted in accordance with the Consolidated Standards Of Reporting Trials (CONSORT) checklist for the RCT design [34]. Participants will be randomly assigned following blind simple randomization procedures when they complete the baseline assessment (randomization sequence under computer control adopting computerized random numbers) to 1 of 2 treatment groups (see Appendix A for the study design and assessments). We only keep the 1:1 ratio constraint at the 260th participant, meaning that when one group reaches 130 participants, all the remaining volunteers (up to the total of 260) will be allocated to the other group. Students will be randomized to IB-MAI or CC. Participants in the intervention condition will receive an 8-week unguided IB-MAI program, i.e. MindBlooming. Participants in the CC condition will access websites with information about their specific areas of distress identified at baseline. Participants will be assessed with repeated measures (baseline, 8 weeks, 12 weeks, and 24 weeks). The RCT participants’ flow is presented in Figure 2.

### 2.4. Intervention

MindBlooming is conceived as an Internet-Based Multi-Approach Intervention (IB-MAI) intended to provide psychoeducation and teach new strategies to cope with different psychological problems. We consider MindBlooming to be multi-approach since it integrates different psychotherapeutic strategies such as Cognitive Behavioral Therapy [35], Acceptance and Commitment Therapy [36], Mindfulness-Based Cognitive Therapy (MBCT) [37,38], Mindfulness-Based Stress Reduction (MBSR) [39], and Dialectical Behavioral Therapy [40]. All these interventions are evidence-based and several studies proved their efficacy in treating different psychological conditions (for an overview, see [41] and APA division 12 list of research-supported psychological treatments, https://div12.org/psychological-treatments/, accessed date 29 December 2021). The intervention consists of 6 interactive modules: depression, anxiety, sleep problems, self-disruptive thoughts, job- and study-related stress and burnout, and chronic pain. A total of two modules will be assigned to each participant based on the initial screening and specific requests and preferences. In addition to the assigned treatments based on the initial screening, all participants will receive a separate training to improve their relational and empathy skills: during the 6th week, they will have access to two videos focused on the importance of relationships on our health, social support as a protective factor, social isolation and loneliness as a risk factor, emotion regulation, and empathy. Afterward, they will have access to a set of exercises, including quizzes about social relationships, assertive communication training, DBT [40], and Interpersonal Psychotherapy (IPT) for Depression strategies (which correlate depressive symptomatology to relational problems) [42]. Moreover, during the 7th week of treatment, more contents in the same formats (videos, texts, and quizzes) will be proposed, focusing on emotions and relationships problems related to the specific issues faced in the assigned modules (e.g., problems in relationships in association with depressive symptoms).

Every week for the entire duration of the intervention, participants will have access to one session per module, except for the 6th week, which is dedicated to the treatment of relational problems: regardless of the specific issue (depression or anxiety, etc.), this training deals with relational issues relevant to everyone, hence the importance of transversally proposing it. The modules will be presented as follows:○1st week: module 1 and 2 sessions;○2nd week: module 1 and 2 sessions;○3rd week: module 1 and 2 sessions;○4th week: module 1 and 2 sessions;○5th week: module 1 and 2 sessions;○6th week: relational problems related to difficulties in emotion regulation and empathy deficits session;○7th week: module 1 and 2 sessions about relapse prevention, future planning by reflecting on how to attain goals and implement behavioral changes, orientation, and conclusion.

All online sessions will include psychoeducational videos (brief seminars of 15 min each, with audios and PowerPoint presentations), about epidemiology, explanatory models, risk and protection factors, treatments, and some specific aspects of each psychological issue. Once the videos have been seen, training exercises will be proposed to help participants to integrate the content of the sessions into their daily lives (Figure 3). Contents and approaches used for each module are further presented in Table 1.

Quizzes will be presented during each session, along with the possibility for the participants to reply and insert text. Subsequent sessions will be available only if the previous session, and a minimum of related exercises, is completed within 7 days (+3 days to be flexible in the case of students unable to maintain the pace). Even if the contents are completed before the 7th day from the start, new contents will be available from the 7th day onwards. When a new week of treatment is available, participants have 3 days to start completing the tasks. If the subjects do not respect the deadlines, they will be excluded from the study. Participants also have access to a list of available supplementary resources.

For the entire duration of the treatment, participants must answer daily and weekly questions related to each of the two treatment modules they are individually following to constantly monitor the progress of some key variables. The sixth week is an exception, as daily and weekly questions will also address relational issues. Answers are saved and shown to participants on a calendar to follow-up changes over time. Moreover, participants are informed that they can contact an e-Coach for technical questions and concerns or emotional crisis at any time during the intervention.

### 2.5. Control Condition

For each participant, the CC consists of indications to websites specifically focused on the different psychological conditions (depression, anxiety, sleep problems, self-disruptive thoughts, job- and study-related stress and burnout, and chronic pain) they report as areas of distress according to the global screening.

### 2.6. Outcome Measures

Participants will be screened with a complete online battery to assess their symptoms, including both primary and secondary outcome measures, at pretreatment baseline (T0), after the 7-week IB-MAI treatment, or CC completion (T1), at 12-week follow-up (T2), and 24-week follow-up (T3) (see Appendix A for the study design and assessments). Moreover, at pretreatment time (T0), socio-demographic questions are included; these concern marital status, educational qualification, employment status, physical or psychiatric illnesses, and past or current therapies. Questions related to gender and the COVID-19 pandemic are included to investigate the influence of the pandemic on students’ mental health status. These items were drawn up according to the International COVID-19 Suicide Prevention Research Collaboration guidelines [56]. In contrast, those relating to gender were taken from the Williams Institute of the UCLA School of Law for health surveys [57].

Since participants have a timeframe to begin and complete each week of treatment, the overall treatment duration can differ among subjects. Therefore, post-treatment and follow-up time points will be calculated for each participant, starting from the end of the last activity.

#### 2.6.1. Primary Outcome Measures

❖Beck Depression Inventory (BDI-II) [58,59], composed of 21 items investigating depressive symptoms (see Appendix A for a thorough description of the scales).❖State-Trait Anxiety Inventory (STAI) [60,61], consisting of 40 items exploring state and trait anxiety.❖Italian Version of the Pittsburgh Sleep Quality Index (PSQI) [62,63], consisting of 19 items to investigate sleep quality in the last month.❖Brief version of the Columbia-Suicide Severity Rating Scale (C-SSRS) [64] modified, consisting of 4 items investigating current suicide risk and 5 items investigating lifetime risk.❖School Burnout Inventory (SBI) [65,66], composed of 9 items exploring three dimensions of burnout, i.e., exhaustion, cynicism, and sense of inadequacy.❖Difficulties in Emotion Regulation Scale (DERS-20) [67,68] consisting of 20 items investigating emotional regulation skills, and the Brief Interpersonal Reactivity Index (B-IRI) [69,70], consisting of 16 items investigating empathic response skills.❖Brief Pain Inventory (BPI) [71,72], composed of 15 items measuring the intensity of pain and its impact on daily functioning.❖World Health Organization Quality of Life—short version (WHOQOL-BREF) [73], consisting of 26 items investigating the quality of life.

#### 2.6.2. Secondary Outcome Measures

❖Visual Analogue Scales (VAS) for psychological and physical pain (6 items) [32].❖Insomnia Severity Index (ISI) to investigate the impact of insomnia in the last two weeks (7 items) [74].❖Toronto Alexithymia Scale (TAS-20), consisting of 20 items investigating levels of alexithymia [75,76].❖Coping Orientation to the Problems Experienced (COPE-NVI-25) [77,78], consisting of 25 items investigating coping strategies.❖Life Events Checklist (LEC) [79], composed of 16 items investigating the subject’s stressful experiences.❖Reasons for Living Inventory (RFLI) [80], consisting of 48 items investigating the subject’s reasons for living.❖COVID-19-related questions compiled according to the International Association for Suicide Prevention (IASP) guidelines for suicide prevention research [56].

Given the feasibility nature of this study, data on the program’s acceptability will also be collected at post-treatment (T1) through the same measures used in the pilot study and described in the related paragraph.

### 2.7. Sample Size Calculation

Considering the attrition rate, we will recruit 260 students to obtain 200 study completers. A total of two-hundred participants will allow to detect an effect size of at least d = 0.40, fixing a power (1 − β) of 80%, and an alpha of 0.05 (two-tailed) in a between group comparison (e.g., comparing the two treatments). This sample size is similar to Harrer et al. [25]. The effect size is estimated considering a recent meta-analytic review for Internet-based stress interventions which reports effect sizes ranging between d = 0.32 and d = 0.64 [81].

Frequentist and Bayesian approaches have their own advantages and drawbacks in data analysis. We will adopt both to provide robust evidence to the specific statistical approach.

We conducted a Bayes Factor Design Analysis (BFDA) [82] (i.e., the counterpart of power analysis in the Bayesian world) to estimate the Bayesian analysis sensitivity, given the sample size of 200 people. We simulated 10,000 results, fixing the Bayes Factor (BF) decision boundary at 10, and an estimated effect size for a true H1 equal to d = 0.40. The BFDA simulation was performed for a default prior (i.e., a Cauchy distribution with a scale = 0.707) and an informative prior (i.e., a t distribution with μ = 0.35, df = 3, r = 0.102), considering true H0 (i.e., d = 0) and H1 (d = 0.40). Given those parameters, we estimate to surpass the decision boundary for H1 in 87.8% adopting the default prior and 93.8% of the case adopting the informed prior. In the case of a true H0, the decision boundary is surpassed in 45.5% and 62.2% of the cases adopting the default or the informed prior, respectively. The design is robust to the misleading evidence, showing a false positive evidence rate of 0.002 and 0.006 adopting the default or the informed prior. No false negative evidence has been observed over 10,000 simulations with any of the two priors. In the case of inconclusive evidence, we will increase the sample size until the BF decision boundary is surpassed (BF > 10) in either one of the two directions supporting H1 or H0.

We expect to have enough data to tune a model that highlights treatment response predictors and different trajectories in students’ response to treatment. We will test five different algorithms: Linear Discriminant Analysis (LDA); Classification and Regression Trees (CART); k-Nearest Neighbors (kNN); Support Vector Machines (SVM) with a linear kernel; and Random Forest (RF). This will offer a good mixture of simple linear (LDA), non-linear (CART, kNN), and complex non-linear methods (SVM, RF).

### 2.8. Statistical Methods

Statistical analyses will be conducted adopting both a frequentist and a Bayesian approach to provide robust evidence favoring or against the intervention.

The main study outcome is to verify the effectiveness of the intervention: this would be verified by a T0 to T1 delta different from 0. Additionally, we will test the effectiveness of the IB-MAI against the CC group comparing the two T0 to T1 deltas. We will use a series of *t*-tests and its Bayesian [83] corresponding test.

Stability over time will be verified with a mixed ANOVA using the T1, T2 and T3 datapoints as within-subject measures and group (IB-MAI versus CC) as a between-subjects factor. We will use “classic” (i.e., frequentist) ANOVA and its corresponding Bayesian version [84] for each primary and secondary outcome measure.

For what concerns the predictive model, we will explore the possibility to use ML algorithms to tune a model able to predict the responders on each of the primary outcomes starting from the socio-demographic information collected and the primary and secondary outcome measures in two steps. First, we will use only socio-demographic information to predict the primary outcome measure of each module. Second, we will use the socio-demographic information, the other primary, and all the secondary outcome measures to predict the primary outcome of each specific module. Before entering an outcome measure as a target variable (i.e., to be classified/predicted variable), variables have to be dichotomized. We will classify participants as responders or non-responders, adopting the minimum clinical importance difference and adopting a distribution approach [85]. A responder will be defined as a participant that records a change larger than 1 standard error of the mean (SEM) calculated on the baseline distribution. For example, the effectiveness of the first module is measured primarily on the BDI-II scale. We will first identify participants that change more than 1 SEM to classify responders and non-responders. Then we will use ML algorithms to predict this outcome starting from socio-demographic in a first step, and socio demographic plus the other outcome variables (except BDI-II of course) in a second step. We will test five different algorithms to have a good mixture of simple linear (LDA), nonlinear (CART, kNN), and complex nonlinear (SVM, RF) methods, thus covering the most used and diffused ML methods [86]. A total of 20% of the data will be held out for testing purposes, thus ML training will be performed on the remaining 80% data of the total sample. The portion of sample holdout will be set at the beginning and kept constant for all the algorithms tested and outcomes targeted, to make the different models’ performances directly comparable.

## 3. Discussion

This paper describes MindBlooming, a new IB-MAI designed to promote mental health in university students, and describes the study protocol, testing its efficacy through a pilot study and a two-arm RCT. The treatment will be compared to a control condition. Participants’ acceptance and usability of the app will also be assessed to identify possible implementation barriers.

A primary limitation of our project might be represented by the treatment dropout level; due to its overall structure and duration, an IB-I such as MindBlooming requires user commitment and the investment of time and resources, which are not always available. Moreover, some common psychological issues were not addressed within the treatment due to their complexity (i.e., eating disorders and substance use disorder). Further limitations concern the assessment phase and the fully automated nature of the treatment. Indeed, only self-report questionnaires are used to identify possible psychological issues and assign modules during the assessment phase. Although this simplifies the procedure, it also makes it more challenging to provide an in-depth assessment. Lastly, the user’s complete autonomy does not make possible to verify whether theoretical sessions of psychoeducation and exercises have been completed.

## 4. Conclusions

The MindBlooming project is a straightforward attempt to respond to the rising levels of psychological distress among college students by providing a readily accessible and always-available treatment. Similar studies have been conducted or are planned to be performed worldwide and became even more essential after the COVID-19 pandemic, but, to the best of our knowledge, this is the first study in Italy with these characteristics (i.e., a multi-approach IBI that deals with different psychological and physical problems, and is personalized). Hopefully, we assume that the results of our study will contribute to the growing research on Internet-delivered treatments. If the expected results are obtained, MindBlooming could contribute to: (a) the development of more adequate treatments for low- to moderate-intensity mental health symptomatology, tailored to individual needs and targeting specific symptoms; (b) the improvement of the quality of life of university students; and (c) the consistent reduction of mental health care system costs. MindBlooming is easily scalable, allowing it to also be implemented in different contexts and populations (e.g., university workers, people with mental disorders, and adolescents).

## Figures and Tables

**Figure 1 ijerph-19-02711-f001:**
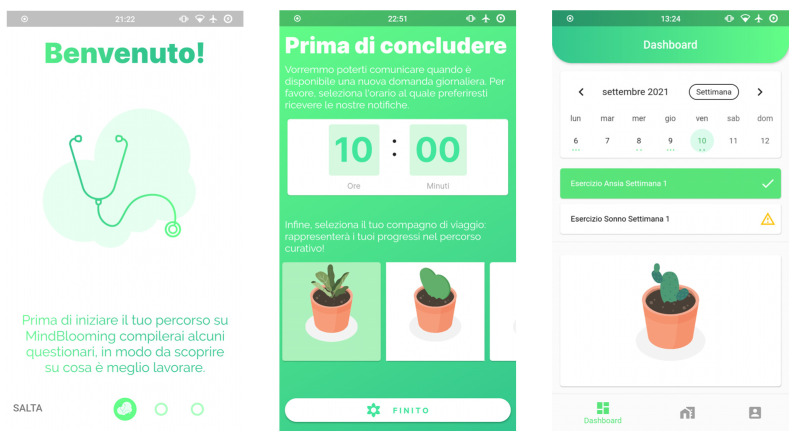
MindBlooming app user interface.

**Figure 2 ijerph-19-02711-f002:**
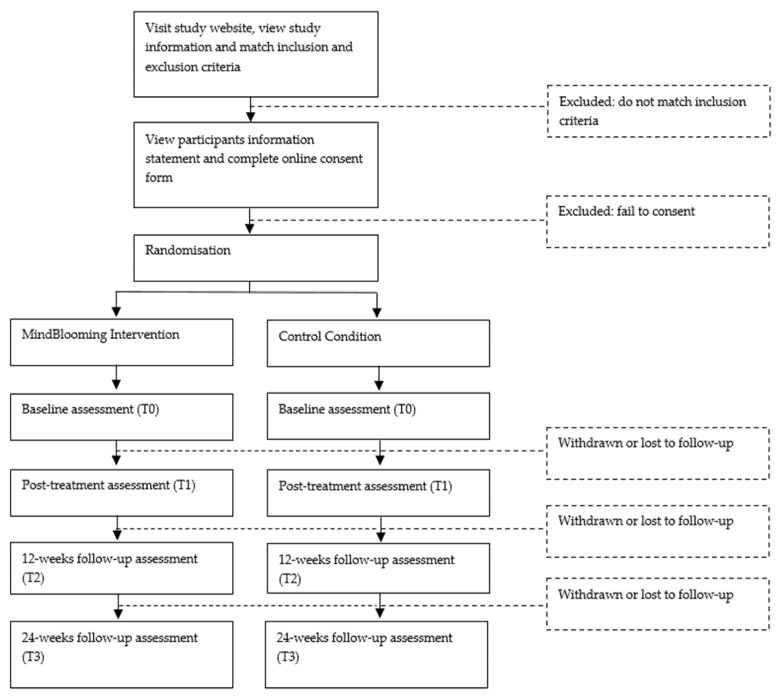
RCT’s participants’ flow.

**Figure 3 ijerph-19-02711-f003:**
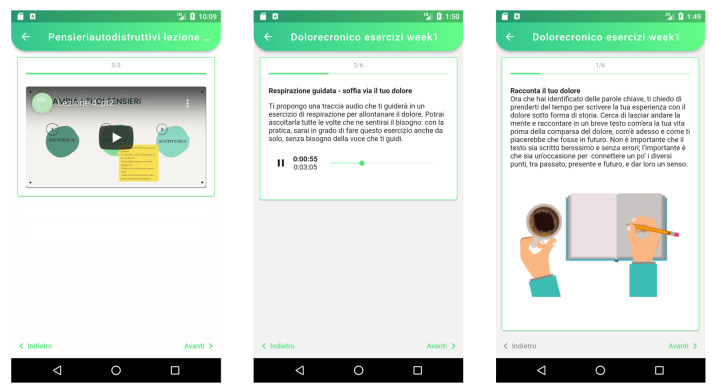
Screenshot of an example of psychoeducational video and exercises.

**Table 1 ijerph-19-02711-t001:** Contents and approaches of each module.

Module	Psychoeducation	Exercises	Approaches
Depression	(1) The definition of depression and its epidemiology; (2) Two psychological models of depression; (3) Recognizing and challenging unhelpful thoughts, emotions, and behaviors; (4) Learning problem-solving strategies; (5) Improving levels of motivation and self-esteem; (6) Preventing and managing relapses.	Thoughts challenging; Problem-solving; Shifting attention; Behavioral activation; Relaxation and breathing strategies; Mindfulness techniques.	❖ Cognitive Behavioral Therapy [35,43]
Anxiety	(1) Definition of anxiety, its epidemiology, and major causes; (2) Psychological models of anxiety and useful strategies for its management; (3) Definition of a particular type of anxiety disorder: panic disorder; (4) Problems related to anxiety, its components, reduction of avoidance, and safety behaviors; (5) Emotions, behavior, and mindfulness techniques; (6) Preventing and managing relapses.	Thoughts challenging; Problem-solving; Shifting attention; Behavioral activation; Relaxation and breathing strategies; Mindfulness techniques.	❖ Cognitive-Behavioral Therapy for Anxiety Disorders [43,44]; ❖ Acceptance and Commitment Therapy (ACT) [36]
Sleep problems	(1) Defining the architecture of normal sleep and its role in daily functioning; (2) The insomnia problem; (3) Cognitive techniques of the Cognitive Behavioral Therapy for Insomnia; (4) Behavioral aspects of Cognitive Behavioral Therapy for Insomnia; (5) Identifying the other sleep disorders; (6) Maintenance strategies and relapse prevention.	Sleep restriction; Breathe and relaxation exercises; Progressive muscles relaxation; Restriction of time in bed; Cognitive restructuring; Thoughts recording and challenging.	❖ Cognitive-Behavioral Therapy for Insomnia (CBT-I) [45,46,47,48]; ❖ Progressive muscle relaxation [49,50]
Self-disruptive thoughts	(1) Definition of self-disruptive thoughts and epidemiology; (2) Psychological models of self-disruptive thoughts and psychotherapeutic strategies; (3) Identifying and challenging negative thoughts; (4) Creating a weekly action plan; (5) Structuring the problem solving; (6) Maintenance strategies and relapse prevention.	Security plan; Identifying life reasons; Building a weekly plan of well-being; Relaxation strategies; Thoughts challenging; Behavioral activation; Problem-solving.	❖ Brief Cognitive-Behavioral Therapy for Suicide Prevention (BCBT) [51]; ❖ Cognitive Behavioral Therapy for Suicide Prevention (CBT-SP) [52,53,54]; ❖ Dialectical Behavior Therapy (DBT) [40,54,55]
Job- and study-related stress and burnout	(1) Definition of stress and burnout, work context, and epidemiology; (2) Stress and burnout in university students and burnout explanatory models; (3) Risk and protective factors and treatment; (4) Coping strategies, resilience, and the importance of emotions; (5) Errors of thought and perfectionism, and procrastination; (6) Recap and maintenance of strategies.	Meditation audios; Relaxational training; Mindfulness exercises; Progressive muscular relaxation.	❖ Mindfulness-Based Stress Reduction (MBSR) [39]; ❖ Reduction Progressive relaxation [49]
Chronic pain	(1) Definition of chronic pain and epidemiology; (2) Psychological models of chronic pain, useful strategies for its management, and goal setting; (3) Challenging unhelpful thoughts and pain acceptance; (4) Coping strategies and pacing; (5) Emotions and assertive communication; (6) Preventing and managing relapses.	Relaxation strategies; Mindfulness audios; Thoughts challenging; Behavioral activation; Pacing; Problem-solving; Assertive communication.	❖ Cognitive Behavioral Therapy (CBT) [43]; ❖ Acceptance and Commitment Therapy (ACT) [36]; ❖ Mindfulness-Based Cognitive Therapy (MBCT) [37,38]

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
