# Peer review of "An Internet-Based Multi-Approach Intervention Targeting University Students Suffering from Psychological Problems: Design, Implementation, and Evaluation"

_ijerph, 2022, doi:10.3390/ijerph19052711_

Round 1

Reviewer 1 Report

The article is interesting for readers, but the scientific reliability is low. The article does not sufficiently explain the used sample, its selection, characteristics of respondents. The discussion is insufficient, the results need to be discussed in more depth with the state of knowledge. I recommend an extensive revision.

Author Response

The article is interesting for readers, but the scientific reliability is low. The article does not sufficiently explain the used sample, its selection, characteristics of respondents. The discussion is insufficient, the results need to be discussed in more depth with the state of knowledge. I recommend an extensive revision.

Reply:

We better described the sample selection.

Reviewer #1 asks us to discuss results in more depth. However, our manuscript describes a study protocol, thus we do not present results in this manuscript. However, we followed the reviewer’s advice (also considering comments from other reviewers and we consistently changed the structure of the discussion, even if, with the absence of the results, being this a protocol, the discussion is still limited.

Reviewer 2 Report

Results should be reported and emphasized in the Summary.

Results should be discussed in the Discussion.

Author Response

Results should be reported and emphasized in the Summary.

Results should be discussed in the Discussion.

Reply:

We are sorry that this point was not clear. Unfortunately, results are still not present because this piece of work is a protocol. We hope to have them soon.

Reviewer 3 Report

This is an innovative and very important document. However, I thought the discussion could be richer.

Author Response

This is an innovative and very important document. However, I thought the discussion could be richer.

Reply:

We consistently improved the discussion, removing parts already present in the introduction. However, with the absence of the results, being this a protocol, the discussion is still limited.

Reviewer 4 Report

The opinions discussed in this article whose title is 'An Internet-Based Multi-Approach Intervention Targeting University Students Suffering from Psychological Problems: Design, Implementation and Evaluation' are innovative and meaningful to practice. However, you have to amend your article meticulously according to the review. Details are as follows.

â‘  Introduction

‘In the European Union (E.U.) one out of six individuals was affected by a mental health issue in 2016 [1,2]. ’

- Please update the latest data(by the year of 2021 or 2022) on the prevalence of mental health issue, that will make your article more advanced.

Despite the psychological problems', high ...

- Please check for punctuation errors. 

‘To the best of our knowledge, this will be the first attempt to deliver such an intervention to the university students' population.’

- Please do more literature searches to confirm if your study is really the first attempt of the same type of intervention.

‘Secondary aims will be: 2) the feasibility assessment of this IB-MAI (in terms of acceptability, satisfaction, usability, and uptake) through a pilot study on a restricted sample; 3) a randomized controlled trial (RCT) to compare the IB-MAI with web-education, as the control condition (CC); 4) exploratory analysis to tune a predictive model on the collected data adopting Machine Learning techniques.’

- The digit numbers of secondary aims are wrong.

‘The project has already been approved by the Internal Review Board (IRB) of the

Milano - Bicocca University at the beginning of 2021. ’

- The institutional review board statement is recommended to put in the ‘Methods’ - section, and please add the protocol code.

- Please check the abstract content and writing standard. You should reduce the narration of unnecessary content.

â‘¡ Materials and Methods

- In general, you should reduce the narrative of unnecessary and repetitive content in this section. It is also necessary to reorganize and reorder the content of each part.

- What theories and application requirements are the functions of your IB-MAI designed based on?

- You need to complete the details of these scales you use in ‘2.1.1. Feasibility measures ’ ‘2.5.1. Primary outcome measures ’ and ‘2.5.2. Secondary outcome measures ’, such as who prepared the scale? What items it includes? How to use it for evaluation and scoring? etc.

- 2.2. RCT design

- It is recommended that you refer to the ‘CONSORT Statement’ to improve the content of the RCT design.

- Did the experiment use a blinding? How can randomization be guaranteed during the whole intervention?

-2.2. Sample size calculation

- There is no need to present the calculation process of the sample size in too much detail, you can delete the redundant words.

- Has the possible dropout rate been taken into account when calculating the minimum sample size?

-2.3. Participants and recruitment

- Shouldn't the inclusion criteria for participants include 'suffering from mental illness'?

- Based on your diverse participants recruitment channels, how to ensure that there is no difference in baseline data? And what about the same standards for data collection?

- 2.4. Intervention

You will need to supplement an introduction to the research background of the relevant theories and methods you used, such as Cognitive Behavioral Therapy, Acceptance and Commitment Therapy, Mindfulness-Based Cognitive Therapy, Mindfulness-Based Stress Reduction, and Dialectical Behavioral Therapy, and detail how you are intervening with these therapies.

- 2.5. Outcome measures

Participants will be screened with a complete online battery to assess their symptoms, including both primary and secondary outcome measures, at pre-treatment baseline (T0), after the 7-week IB-MAI treatment, or CC completion (T1), at 12-week follow-up (T2), and 24-week follow-up (T3) (see Figure 1S for the study design).

- What is the rationale for developing the four evaluation times?

- Why is there no the third part (3.) between ‘2. Materials and Methods’ and ‘4. Discussion’ of this article?

â‘¢ Discussion

- In this section you should focus on what is relevant to the design of this intervention study rather than repeating the content of ‘Introduction’ section. 

MindBlooming will hopefully contribute to: a) the development of more adequate treatments for low-intensity mental health symptomatology, tailored to individual needs and targeting specific symptoms; b) the improvement of the quality of life of university students; c) the consistent reduction of mental health care system costs. MindBlooming is easily scalable, allowing to implement it also in different contexts and populations (e.g., university workers, people with mental disorders, adolescents).

- It is unreliable to speculate on the effectiveness of this scheme based on the results of this research, as this intervention study has not been conducted.  

- The above content should belong to the ‘conclusion’ part and not the ‘discussion’.

Author Response

The opinions discussed in this article whose title is ‘An Internet-Based Multi-Approach Intervention Targeting University Students Suffering from Psychological Problems: Design, Implementation and Evaluation’ are innovative and meaningful to practice. However, you have to amend your article meticulously according to the review. Details are as follows.

â‘  Introduction

‘In the European Union (E.U.) one out of six individuals was affected by a mental health issue in 2016 [1,2]. ’

- Please update the latest data(by the year of 2021 or 2022) on the prevalence of mental health issue, that will make your article more advanced.

Reply:

Thank you for this suggestion. We changed the reference to more recent data.

Despite the psychological problems’, high ...

- Please check for punctuation errors.

Reply:

We agree with the suggestion and changed the punctuation in the indicated sentence.

‘To the best of our knowledge, this will be the first attempt to deliver such an intervention to the university students’ population.’

- Please do more literature searches to confirm if your study is really the first attempt of the same type of intervention.

Reply:

We performed a further literature search.

Please see:

Taylor, C. B., Graham, A. K., Flatt, R. E., Waldherr, K., & Fitzsimmons-Craft, E. E. (2021). Current state of scientific evidence on Internet-based interventions for the treatment of depression, anxiety, eating disorders and substance abuse: an overview of systematic reviews and meta-analyses. European journal of public health, 31(Supplement_1), i3-i10.

Huang, J., Nigatu, Y. T., Smail-Crevier, R., Zhang, X., & Wang, J. (2018). Interventions for common mental health problems among university and college students: A systematic review and meta-analysis of randomized controlled trials. Journal of psychiatric research, 107, 1-10.

Reins, J. A., Buntrock, C., Zimmermann, J., Grund, S., Harrer, M., Lehr, D., ... & Ebert, D. D. (2021). Efficacy and moderators of internet-based interventions in adults with subthreshold depression: an individual participant data meta-analysis of randomized controlled trials. Psychotherapy and Psychosomatics, 90(2), 94-106.

With this sentence, we meant that “To the best of our knowledge, this will be the first attempt to deliver such an intervention - that is IBI, multi-approach, dealing with different psychological and physical problems, and personalized - to the university students’ population.” We added these specifications at the end of the introduction.

‘Secondary aims will be: 2) the feasibility assessment of this IB-MAI (in terms of acceptability, satisfaction, usability, and uptake) through a pilot study on a restricted sample; 3) a randomized controlled trial (RCT) to compare the IB-MAI with web-education, as the control condition (CC); 4) exploratory analysis to tune a predictive model on the collected data adopting Machine Learning techniques.’

- The digit numbers of secondary aims are wrong.

Reply:

We changed the numbers of secondary aims to make them clearer.

‘The project has already been approved by the Internal Review Board (IRB) of the Milano - Bicocca University at the beginning of 2021. ’

- The institutional review board statement is recommended to put in the ‘Methods’ - section, and please add the protocol code.

Reply:

Thank you for the revision, we have added the protocol code and moved the content to the “Methods” section.

- Please check the abstract content and writing standard. You should reduce the narration of unnecessary content.

Reply:

We are very sorry, but we checked the abstract content and writing standard and we were not able to find unnecessary and repetitive contents (word count: 202). We changed a not very clear word but for the rest we really believe that all points are important for the explanation of the project. We hope that the reviewer might agree on this point.  

â‘¡ Materials and Methods

- In general, you should reduce the narrative of unnecessary and repetitive content in this section. It is also necessary to reorganize and reorder the content of each part.

Reply:

We tried to reduce this section and to reorganize the content of each paragraph.

- What theories and application requirements are the functions of your IB-MAI designed based on?

Reply:

We followed the design principle of Material Design [1], which is a system of guidelines, components, and tools that support the best practices of user interface design.

We added this point in Materials and Methods paragraph.

 [1] https://material.io/design

- You need to complete the details of these scales you use in ‘2.1.1. Feasibility measures ’ ‘2.5.1. Primary outcome measures ’ and ‘2.5.2. Secondary outcome measures ’, such as who prepared the scale? What items it includes? How to use it for evaluation and scoring? etc.

Reply:

Considering the already high number of words of the manuscript, we believe it is better to avoid to add so many details for each scale. We indicated only the total number of items, what the scale measures and the authors who validated it both in Italian and original version. We hope you may agree on that.

- 2.2. RCT design

- It is recommended that you refer to the ‘CONSORT Statement’ to improve the content of the RCT design.

Reply:

We added this point in the “2.3 RCT design” paragraph.

- Did the experiment use a blinding? How can randomization be guaranteed during the whole intervention?

Reply:

We added this sentence to the “2.3 RCT design” paragraph: “Participants will be randomly assigned following blind simple randomization procedures when they will complete the baseline assessment”.  

-2.2. Sample size calculation

- There is no need to present the calculation process of the sample size in too much detail, you can delete the redundant words.

Reply:

Thank you for the suggestion. However, considering that this paper contains the experimental protocol, we believe that it is crucial to define how we established the sample size and the related analytical procedures. Nonetheless, we followed the reviewer’ suggestion to reduce the form. We have removed the unnecessary content and rephrased the paragraph more concisely.

- Has the possible dropout rate been taken into account when calculating the minimum sample size?

Reply:

We added this sentence to the “2.6 Sample size calculation” paragraph: “Considering the attrition rate, we will recruit 260 students to obtain 200 study completers”.  

-2.3. Participants and recruitment

- Shouldn’t the inclusion criteria for participants include ‘suffering from mental illness’?

Reply:

We completely agree with this point. We better specified it in the paragraph “2.1 Participants and recruitment”. To enter the study, we mistakenly did not clearly state that there must be (at least) a mild to moderate intensity in each participant’s symptomatology: this inclusion criteria is not a dichotomous variable of mental issues’ absence/presence, but a degree of psychological problems that must be at least mildly-moderately present.

- Based on your diverse participants recruitment channels, how to ensure that there is no difference in baseline data? And what about the same standards for data collection?

Reply:

We added this information in the paragraph “2.1 Participants and recruitment” (“We will collect the information concerning the diverse participants’ recruitment channels to ensure that there is no difference in baseline data among different recruitment channels. The same standards for data collection will be applied through different channels.”).

- 2.4. Intervention

You will need to supplement an introduction to the research background of the relevant theories and methods you used, such as Cognitive Behavioral Therapy, Acceptance and Commitment Therapy, Mindfulness-Based Cognitive Therapy, Mindfulness-Based Stress Reduction, and Dialectical Behavioral Therapy, and detail how you are intervening with these therapies.

Reply:

We added this specification and references related to the research support of the psychotherapeutic strategies used in the paragraph “2.4 Intervention”. Further details are presented in Table 1.

- 2.5. Outcome measures

Participants will be screened with a complete online battery to assess their symptoms, including both primary and secondary outcome measures, at pre-treatment baseline (T0), after the 7-week IB-MAI treatment, or CC completion (T1), at 12-week follow-up (T2), and 24-week follow-up (T3) (see Figure 1S for the study design).

- What is the rationale for developing the four evaluation times?

Reply:

The rationale behind the four evaluation times is to provide a more accurate assessment not only of the pre- and post-treatment but also of a period of 3-6 months after the treatment, to assess the duration of the potential improvements and to delineate trajectories of response.

- Why is there no the third part (3.) between ‘2. Materials and Methods’ and ‘4. Discussion’ of this article?

Reply:

Thank you for the point, it was a mistake. We corrected it.

â‘¢ Discussion

- In this section you should focus on what is relevant to the design of this intervention study rather than repeating the content of ‘Introduction’ section.

Reply:

We consistently changed the Discussion.

MindBlooming will hopefully contribute to: a) the development of more adequate treatments for low-intensity mental health symptomatology, tailored to individual needs and targeting specific symptoms; b) the improvement of the quality of life of university students; c) the consistent reduction of mental health care system costs. MindBlooming is easily scalable, allowing to implement it also in different contexts and populations (e.g., university workers, people with mental disorders, adolescents).

- It is unreliable to speculate on the effectiveness of this scheme based on the results of this research, as this intervention study has not been conducted. 
- The above content should belong to the ‘conclusion’ part and not the ‘discussion’.

Reply:

We agree with the revisions and we have moved the paragraph to the conclusion.

Round 2

Reviewer 1 Report

I recommend accepting the article.

Author Response

Many thanks for this positive feedback.

Reviewer 4 Report

The opinions discussed in this article whose title is 'An Internet-Based Multi-Approach Intervention Targeting University Students Suffering from Psychological Problems: Design, Implementation and Evaluation' are innovative and meaningful to practice. Your careful revisions make this article look much better now than the previous version, but there are still some issues that need to be fixed further. Details are as follows. â‘  Materials and Methods - What theories and application requirements are the functions of your IB-MAI designed based on? Have the specific contents of the intervention APP been reviewed by relevant experts to ensure their rationality? - How were the included participants determined to have an associated mental illness in ‘2.1 Participants and recruitment’? Please describe your diagnosis. - If participants suffer from mental illnesses other than the six mentioned in the article, are they allowed to participate in the program? - On what basis did you designate the two intervention modules for each participant in ‘2.4. Intervention’. - Please describe the measures implemented in the control group(CC condition) during the intervention. - You need to complete the details of these scales you use in ‘2.1.1. Feasibility measures ’ ‘2.5.1. Primary outcome measures ’ and ‘2.5.2. Secondary outcome measures ’. If you are concerned about the length of the article, you may consider presenting the research questionnaire and scales as an appendix file.

Author Response

The opinions discussed in this article whose title is 'An Internet-Based Multi-Approach Intervention Targeting University Students Suffering from Psychological Problems: Design, Implementation and Evaluation' are innovative and meaningful to practice.

Reply:

We thank the reviewer for her/his generally positive evaluation.

Your careful revisions make this article look much better now than the previous version, but there are still some issues that need to be fixed further. Details are as follows.

â‘  Materials and Methods - What theories and application requirements are the functions of your IB-MAI designed based on?

Reply:

The user experience has been designed according to the material design that specifies theories and practical applications of the interaction among human and machine for mobile development.

We further specified this point in Materials and Methods.

Have the specific contents of the intervention APP been reviewed by relevant experts to ensure their rationality?

Reply:

RC, psychotherapist, supervised the entire work.

How were the included participants determined to have an associated mental illness in ‘2.1 Participants and recruitment’? Please describe your diagnosis.

Reply:

The participants must simply declare to have experienced at least mild-/moderate-intensity forms of mental health symptoms through a yes/no question. No diagnosis will be assigned. We better specified this point in the related paragraph.

If participants suffer from mental illnesses other than the six mentioned in the article, are they allowed to participate in the program?

Reply:

Only participants screened positive to 2 of the six psychological problems will receive the treatment. The others, if interested, will receive the treatment without being part of the trial.

On what basis did you designate the two intervention modules for each participant in ‘2.4. Intervention’.

Reply:

Participants will receive the intervention in the case they will screen positive to 2 related primary outcome measures.

Please describe the measures implemented in the control group(CC condition) during the intervention.

Reply:

We added the description of the CC condition as a seprate paragraph.

You need to complete the details of these scales you use in ‘2.1.1. Feasibility measures ’ ‘2.5.1. Primary outcome measures ’ and ‘2.5.2. Secondary outcome measures ’. If you are concerned about the length of the article, you may consider presenting the research questionnaire and scales as an appendix file.

Reply:

We added Appendix 1.